# Identification and Functional Characterization of General Odorant Binding Proteins in *Orthaga achatina*

**DOI:** 10.3390/insects14030216

**Published:** 2023-02-22

**Authors:** Yu Ma, Yu Li, Zhi-Qiang Wei, Jing-Hao Hou, Yu-Xiao Si, Jin Zhang, Shuang-Lin Dong, Qi Yan

**Affiliations:** Key Laboratory of Integrated Management of Crop Disease and Pests, Ministry of Education/Department of Entomology, Nanjing Agricultural University, Nanjing 210095, China

**Keywords:** *Orthaga achiatina*, general odorant binding protein, fluorescence competitive binding assay, molecular docking

## Abstract

**Simple Summary:**

*Orthaga achatina* (Lepidoptera: Pyralidae) is one of the most serious pests of camphor trees. Insect olfaction play a crucial role in identification of host plants and oviposition sites. In this study, we identified the binding affinities of GOBPs to camphor volatiles and sex pheromone components using fluorescence competition binding assays. Moreover, key amino acid residues that bind to plant volatiles were identified in GOBPs using 3-D structure modeling and ligand molecular docking, predicting the interactions between the GOBPs and the host plant volatiles. Overall, OachGOBP2 showed a wider odorant binding spectrum and higher binding capacity than GOBP1.

**Abstract:**

The olfactory system in insects are crucial for recognition of host plants and oviposition sites. General odorant binding proteins (GOBPs) are thought to be involved in detecting odorants released by host plants. *Orthaga achatina* (Lepidoptera: Pyralidae) is one of the most serious pests of camphor trees, *Cinnamomum camphora* (L.) Presl, an important urban tree species in southern China. In this study, we study the GOBPs of *O. achatina*. Firstly, two full-length GOBP genes (*OachGOBP1* and *OachGOBP2*) were successfully cloned according to transcriptome sequencing results, and real-time quantitative PCR measurements showed that both GOBP genes were specifically expressed in the antennae of both sexes, proposing their important roles in olfaction. Then, both GOBP genes were heterologous expressed in *Escherichia coli* and fluorescence competitive binding assays were conducted. The results showed that OachGOBP1 could bind Farnesol (*K*_i_ = 9.49 μM) and *Z*11-16: OH (*K*_i_ = 1.57 μM). OachGOBP2 has a high binding affinity with two camphor plant volatiles (Farnesol, *K*_i_ = 7.33 μM; α-Phellandrene, *K*_i_ = 8.71 μM) and two sex pheromone components (Z11-16: OAc, *K*_i_ = 2.84 μM; Z11-16: OH, *K*_i_ = 3.30 μM). These results indicate that OachGOBP1 and OachGOBP2 differ in terms of odorants and other ligands. Furthermore, key amino acid residues that bind to plant volatiles were identified in GOBPs using 3-D structure modeling and ligand molecular docking, predicting the interactions between the GOBPs and the host plant volatiles.

## 1. Introduction

The olfactory system is extensively used in insects and plays an essential role in many behaviors, such as feeding, mating, and egg-laying site selection. Odorant binding proteins (OBPs) are the first to interact with odor molecules during olfaction [1,2] and protect odor from degradation by means of odor-degrading enzymes (ODE) during transportation [3]. In Lepidopteran species, OBPs can be classified into three categories based on their amino acid sequence homology: pheromone binding proteins (PBPs), general odorant binding proteins (GOBPs) and others. GOBPs and PBPs are two specific subclasses of OBPs based on similarities in their amino acid sequence and sensilla distribution pattern [4]. The molecular structure of GOBP is highly conserved. The molecular mass of GOBP is 16.0–17.2 kDa, and the isoelectric point is 4.9–5.1 [5]. The sequence characteristics of GOBP2 are similar to those found in previous studies [6]. The expression of GOBP varies in different tissues and at different times. In general, the expression level of GOBP is highest in adult antennae, but the expression of GOBP differs between female and male antennae [7]. In recent years, an increasing amount of research has revealed the ligands of GOBP in many species. For example, the GOBP of *Agrotis ipsilon* was specifically expressed in the antennae, and further fluorescence competitive binding assays showed that GOBP had a strong affinity with β-Caryophyllene and other plant volatiles [8]. The GOBP of *Chilo suppressalis* was explicitly expressed in the antennae of both sexes of moths and had a strong binding energy to some host plant volatiles [9,10]. With the development of insect genome and transcriptome sequencing technologies, more GOBPs have been cloned and identified. The functional studies of GOBP have become more advanced, and many experiments have shown that GOBP can bind not only general odorants but also sex pheromone components [8,11,12]. This indicates that GOBP plays a crucial role in the olfactory function of moths.

*Orthaga achatina* Butler (Lepidoptera: Pyralidae) is a serious pest of camphor trees in China. With the acceleration of urbanization and the increase in camphor planting areas, its incidence is increasing. Plant volatiles plays an essential role in the host selection and egg laying of phytophagous insects. Identifying these active plant volatiles and their application to the behavioral regulation of pests has good application prospects [13]. Numerous studies have shown that GOBPs play a vital role in the perception of plant volatiles [14,15,16,17]. Previously, studies have found that GOBP2 in *O. achatina* has a highly binding affinity with two plant volatiles and two pheromone components [6]. To explore the functions of GOBPs in *O. achatina*, cDNA sequences of GOBP1 and GOBP2 were cloned and their tissue-sex expression profiles were determined by RT-PCR and qPCR analyses. Next, after the two GOBPs had been expressed in *Escherichia coli* cells, their binding properties to plant volatiles and sex pheromone components were determined using a fluorescence competition binding assay. Finally, the key amino acid residues of two GOBP interactions with odors were analyzed using 3-D structure modeling and ligand molecular docking.

## 2. Materials and Methods

### 2.1. Insect Rearing and Tissue Collection

In June 2020, the nymphs of *O. achatina* were collected from Songjiang area of Shanghai and raised in an insect rearing box (29 cm × 20 cm × 16 cm). Fresh camphor leaves were added daily until the larvae pupate. A colony was established in the laboratory at 27 ± 1 °C and 60 ± 5% relative humidity under a 14L: 10D cycle. The newly emerged adults were reared with 10% honey water.

The antennae, heads, thoraxes, abdomens, legs and wings of male and female moths were collected using liquid nitrogen when the 3-day old virgin moths entered the dark stage for 5-6 h. All tissues were stored at −80 °C.

### 2.2. RNA Extraction and cDNA Synthesis

Total RNA was extracted from collected tissues using Trizol Reagent (Invitrogen, Carlsbad, CA, USA) following the manufacturer’s instructions. The quality of RNA was checked using a NanoDrop-2000 (Thermo Scientific, Waltham, MA, USA). cDNA was synthesized using PrimeScript™ RT Master Mix (TaKaRa, Liaoning, China) according to the manufacturer’s instructions.

### 2.3. Cloning, Expression and Purification of Recombinant Proteins

Using transcriptome data already held in our laboratory, we screened using homologous blasting and successfully cloned two full-length cDNA of *O. achatina* GOBPs using PCR. The PCR product was purified from an agarose gel, ligated into the *pEASY-Blunt*3 vector (TransGen Biotech, Beijing, China), and then transferred into *Trans-T*1 cells. The expression and purification of recombinant proteins were conducted using previously described methods [18].

### 2.4. Phylogenetic Tree

GOBPs from 15 Lepidoptera moth species (Appendix A) were selected for tree construction. The N-terminal signal peptides were predicted using the online SignalP-5.0 Server http://www.cbs.dtu.dk/services/SignalP/, accessed on 7 May 2022. The GOBP sequences without signal peptide were aligned using the ClustaLW program and a neighbor-joining tree was constructed using Mega-X under the Jones-Taylor-Thornton amino acid substitution model with pairwise deletion of gaps and 1000 bootstrap replicates [19]. The constructed tree was modified and observed using FigTree 1.4.3 (http://tree.bio.ed.ac.uk/software/figtree/, accessed on 23 May 2022).

### 2.5. RT-PCR and qPCR

The reaction system for RT-PCR was: 12.5 μL of Taq DNA Polymerase, 9.5 μL of nuclease-free water, 1 μL of each forward and reverse primers (10 μM) (Appendix A), and 1 μL of cDNA sample (2.5 ng/μL). The PCR conditions were: 95 °C for 3 min; 31 cycles of 95 °C for 15 s, 55 °C for 15 s and 72 °C for 30 s; and final extension at 72 °C for 5 min. The PCR products were detected using 1.0% agarose gel electrophoresis.

The qPCR specific primers (Appendix A) were designed by Beacon Designer 8.0 (PRIMER Biosoft International, San Francisco, CA, USA). The qPCR was performed with SYBR Premix Ex Taq™ (TaKaRa, Liaoning, China) and QuantStudio™ 6 Flex Real-Time PCR System (Applied Biosystems, Foster City, CA, USA). The qPCR reaction was performed in 20 μL: containing 10 μL of SYBR Green qPCR Master Mix, 0.4 μL of ROX Reference Dye II, 0.4 μL of each primer (10 μM), 1 μL of cDNA (10 ng/μL) template, and 7.8 μL of nuclease-free water. The reaction procedure was same as that used in previous studies [18]. GAPDH and β-actin (Appendix A) were used as the reference gene to normalize the target gene expression (GenBank KT361883) [20]. For each tissue, three biological replications were measured with three technical replicates for each replicate. Gene expression levels were analyzed using the 2^−ΔCT^ method [21].

### 2.6. Fluorescence Competitive Binding Assay

A fluorescence competition binding assay was used to test the binding properties of GOBPs to 22 plant volatiles of camphor [22,23] and 4 sex pheromones of *O. achatina* [24]. The fluorescence competitive binding assay was conducted on an F-7000 fluorescence spectrophotometer with a 1 cm light path quartz cuvette and 10 nm slits for excitation and emission. The excitation wavelength was 337 nm, and the emission spectrum was recorded between 380 and 460 nm. Firstly, the binding characteristics of 1-NPN to protein were detected. We dissolved 2 µM protein in 50 mM Tris HCl (pH = 7.4), then gradually added 1 mM 1-NPN to make its concentration reach 2, 4, 6, 8, 10, 12, 14, 16 and 20 µM, recorded the fluorescence value, and then calculated the binding constant between 1-NPN and the recombinant protein using the Scatchard equation.

When studying the binding constant between odorant and recombinant protein, 1-NPN was used as a fluorescence reporter. We dissolved 2 µM protein and 2 µM 1-NPN in 50 mM Tris HCl, and then gradually added the ligand. All the ligands were diluted in methanol (AR, content greater than 99.5%). The concentration of pheromone component was 0.5–10 µM and the concentration of plant volatile was 1–32 µM. In order to determine the binding concentration of protein, the free radical concentration was plotted with the fluorescence intensity of the maximum emission spectrum. It was assumed that the protein activity was 100% and the binding ratio of protein to ligand was 1:1 in the saturated state. The curve was linearized by the Scatchard method. Then, the concentration of competitive binding ligand was calculated according to the IC_50_ value (the concentration when competitive ligand can replace 50% 1-NPN). The calculation formula was: *K*_i_ = [IC_50_]/(1 + [1-NPN]/K_1-NPN_), where [1-NPN] is the concentration of unbound 1-NPN and K_1-NPN_ is the dissociation concentration of protein/1-NPN complex.

### 2.7. Homology Modeling and Molecular Docking

The amino acid sequence after the removal of signal peptide was compared with the NCBI PDB database (https://blast.ncbi.nlm.nih.gov/Blast.cgi, accessed on 8 July 2022). Among them, the sequence similarities of OachGOBPs with BmorGOBP2 (ID: 2WCK) were the highest. As a result, BmorGOBP2 was chosen as the template for 3-D structure modeling. Homology modeling was carried out using SWISS-MODEL (https://s-wissmodel.expasy.org/, accessed on 8 July 2022). AutoDock Vina version 1.1.2 (http://autodock.scripps.edu, accessed on 15 July 2022) [25] was used to conduct molecular docking to determine the mode of the binding of ligands to OachGOBP1 and OachGOBP2. AutoDock Tools version 1.5.6 [26] was used to generate the docking input files. PyMOL version 2.1.0 (http://www.pymol.org/, accessed on 27 July 2022) was subject to visual analysis using. The molecular docking and visual analysis methods were the same those used by Zhang et al. [16]. In addition, the contact 2D-plots of the protein–ligand interactions are shown using the online website https://plip-tool.biotec.tu-dresden.de/, accessed on 4 February 2023.

### 2.8. Data Analysis

Data (mean ± SE) from various samples (*K*_i_ values) were assessed with one-way nested analysis of variance (ANOVA) followed by a Tukey test for mean comparison. Two-sample analysis was performed by a Student *t-*test using IBM SPSS statistical 21.0 software (SPSS Inc., Chicago, IL, USA).

## 3. Results

### 3.1. Phylogenetic and Sequence Analyses of OachGOBPs

The open reading frame (ORF) lengths of *OachGOBP1* and *OachGOBP2* were 501 bp and 489 bp, respectively. They all included seven conserved cysteines, which are typical features of GOBPs. The amino acid sequence comparison showed that OachGOBP1 had 89%, 79%, 78%, 75% and 70% sequence similarity with AtraGOBP1 (ACX47893.1), GmolGOBP1 (AFH02841.1), BmorGOBP1 (NP_001037496.1), EoblGOBP1 (ACN29680.1) and ObruGOBP1 (KOB67878.1), respectively (Figure 1A). OachGOBP2 was found to be similar to AperGOBP2 (CAA65575.1), SinsGOBP2 (QLI62031.1), OnubGOBP2 (BBB15969.1), AtraGOBP2 (ACX47894.1) and CsupGOBP2 (ACJ07120.1), with 87%, 85%, 84%, 81% and 78% sequence similarity, respectively (Figure 1B).

A phylogenetic tree was constructed for GOBP1 and GOBP2 of 16 Lepidoptera species, including *O. achatina*. The results showed that OachGOBP1 was clustered in a small branch with the AtraGOBP1 and GmolGOBP1, and OachGOBP1 had the highest homology with the AtraGOBP1 and GmolGOBP1 in the previous section. OachGOBP2 was clustered with AperGOBP2 and OnubGOBP2 in a single branch, with the closest genetic distance and the same amino acid sequence similarity results as in the previous section. GOBP1 and GOBP2 were divided into two distinct branches with no intersection (Figure 1C).

### 3.2. Tissue Expression Analysis

The RT-PCR results showed that OachGOBP1 and OachGOBP2 were highly expressed in the antennae (Figure 2A). Furthermore, the tissue expression profiles of OachGOBP1 and OachGOBP2 were determined by qPCR. We found that both OachGOBP1 and OachGOBP2 were highly expressed in the antennae, and were barely expressed in other tissues (Figure 2B,C). In addition, the expression of OachGOBP1 and OachGOBP2 were not significantly different between the male and female antennae.

### 3.3. Fluorescence Binding Assays

Next, two OachGOBPs were heterologous expressed in *E. coli.* Following ultrasonication of the cells, two GOBPs were indicated in the inclusion bodies. The molecular weights of the two recombinant GOBPs were approximately 22 kDa after purification by Ni-NTA. Later, we used enterokinase to remove His-tag and the two GOBPs were approximately 19 kDa, consistent with the predicted molecular weight of 19.45 and 18.60 kDa (Figure 3).

The binding affinity of OachGOBP1 and GOBP2 to the fluorescent probe 1-NPN was first determined. The dissociation constants (K_1-NPN_) were 9.946 and 4.910 μM, respectively (Figure 4), suggesting that 1-NPN could be used for subsequent competitive ligand binding assays. Using 1-NPN as a probe, the relative binding affinities of 26 chemicals, including 4 sex pheromones and 22 plant volatiles of camphor, were measured. The results showed that OachGOBP1 had the most significant binding affinities with the camphor volatile Farnesol (*K*_i_ = 9.49 μM), and the *O.achatina* sex pheromone *Z*11-16: OH (*K*_i_ = 1.57 μM). Meanwhile, OachGOBP2 had a binding ability with 8 plant volatiles [Z3-Hexenyl acetate (*K*_i_ = 14.30 μM), Z3-Hexen-1-ol (*K*_i_ = 18.18 μM), β-Ionone (*K*_i_ = 15.51 μM), Farnesol (*K*_i_ = 7.33 μM), α-Farnesene (*K*_i_ = 13.42 μM), α-Phellandrene (*K*_i_ = 8.71 μM), Linalool (*K*_i_ = 12.05 μM), Camphor (*K*_i_ = 17.49 μM)] and two sex pheromones [*Z*11-16: OH (*K*_i_ = 3.30 μM), Z11-16: Ald (*K*_i_ = 2.84 μM)] (Figure 4).

### 3.4. Protein Structure Homology Modeling and Molecular Docking

Sequence alignment showed that OachGOBP1 and OachGOBP2 had 56% and 76% amino acid homology with BmorGOBP2. The highest identity value in the database indicated that BmorGOBP2 was a good protein template for analyzing the structure of OachGOBPs. Therefore, we used computational programs to obtain the 3-D protein structures of two OachGOBPs based on homologous protein modeling. The results of the structural comparison (Figure 5) show that: (1) OachGOBPs have seven α-helixes (α1-α7), which is the same as Bmor GOBP2 and other moths’ GOBPs [15,16,22,27]; (2) The lumen structure of OachGOBPs is similar to that of BmorGOBP2. Therefore, OachGOBPs and BmorGOBP2 should have similar ligand binding mechanisms, and BmorGOBP2 is a reliable model.

According to the results of the fluorescence competition binding assays, the ligands with binding force to GOBPs were selected for computer simulations. The results showed that the binding energies of all the ligands were negative. At the same time, the amino acid residues around the GOBP-ligands complexes were mostly hydrophobic (Figure 6 and Figure 7), suggesting a strong interaction between OachGOBPs and the plant volatiles. Moreover, some amino acids appeared to be crucial for the ligand binding, and we further analyzed the ligand hydrogen-bonded to the protein. We used PyMol to calculate the distance between each atom of the ligand and carbon atoms on the amino acid residues in the surrounding 4 Å, and the results are shown in Appendix A. The 2D-plots predicting hydrogen bonding results are consistent with the molecular docking results (Appendix A). Remarkably, in the OachGOBP1/Farnesol complexes, the length of hydrogen bonds interacting with Thr32 and Trp60 was as low as 2.1 and 2.3 Å, respectively. Moreover, in the OachGOBP2/Linalool complexes, the length of the hydrogen bonds interacting with Thr31 and Trp59 were as low as 2.0 and 2.3 Å, respectively (Appendix A).

## 4. Discussion

Antennae play an important role in recogniting host plant volatiles by moths. GOBP is widely found in the antennae of moths and is involved in the perception of host plant volatiles. In an in-depth study of the molecular mechanisms of smell in insects, the principle of reverse chemical ecology was used by identifying the GOBP base. The binding ability of different compounds and GOBP was determined to screen the active odor components. Then this substance provides an important basis for further development of attractant or repellent. Phylogenetic trees can analyze the evolutionary process of species and reveal the relationship of genes among different species, and have been widely used to study insect OBP. In this study, the phylogenetic tree of GOBPs from 16 Lepidopteran species revealed that GOBP1 and GOBP2 were clustered separately (Figure 1C), corroborating the situation found in other studies [8,22]. This may be due to the high amino acid sequence similarity of the same type of GOBP among different species of Lepidoptera and the low sequence similarity between two GOBPs of the same species [15,23]. This suggests that both GOBP1 and GOBP2 are conserved in moths, and the same type of GOBPs of different insects may have similar functions. In addition, OachGOBP1 and OachGOBP2 had the highest sequence similarity and the closest genetic distance to different species of insects, respectively (Figure 1), which may be related to the fact that GOBP2 and GOBP1 are chromosomally distant in Lepidoptera. Based on the phylogeny tree, we suggest that OachGOBP1 has a similar function to AtraGOBP1 [28], which was verified in the fluorescence competition binding assay. OachGOBP2 has a similar function to AperGOBP2, although this was not reported.

In contrast to previous results that OachGOBP2 exhibited binding affinity (IC_50_ < 32 μM) to 8 of 19 tested volatile chemicals from camphor trees [6]. Here, the result show that GOBP2 did not bind to Z3-hexenal and Camphene, but it did, however, bind to β-Ionone (*K*_i_ = 15.51 μM) and Camphor (*K*_i_ = 17.49 μM). β-ionone and Camphor were not tested in Liu’s experiment. β-Ionone is a class of cyclized isoprenoids, which are widely distributed in plants. OBPs in multiple insects can detect β-Ionone, such as GOBP2 of *Loxostege sticticalis* [10], OBP21 of *Apolygus lucorμm* [29] and OBP4 of *Adelphocoris lineolatus* [30], indicating that β-Ionone might be a common signal for host plant selection in insects. In addition to detecting plant volatiles, OachGOBP2 has a moderate binding capacity to *Z*11-16: OAc (*K*_i_ = 2.84 μM), the major pheromone component of *O. achatina*. The OachGOBP1 and OachGOBP2 showed strong or moderate binding ability to *Z*11-16: OH, a minor sex pheromone component of *O. achatina* [24]. The binding ability of GOBP to sex pheromone components has been reported in many moths, such as *Laphygma exigua*, *Spodoptera litura* [15], *Chilo suppressalis* [9], and *A. ipsilon* [8]. The AtraGOBP1, which is highly homologous to OachGOBP1, can also bind with the sex pheromone component *Z*11, *Z*13-16: OH of *Amyelosis transitella* [28]. Recently, the function of GOBP2 was confirmed in vivo by using CRISPR/Cas9, demonstrating that the SlitGOBP2 mutant *S. litura* females showed significantly hampered EAG responses and oviposition preference to host plants [31]. In field trapping experiments, the addition of Z11-16: OH to the major component did not change the capture of males [24], suggesting that this component may have other functions. Overall, OachGOBP2 showed a wider odorant binding spectrum and higher binding capacity than GOBP1, which could be explained by the larger hydrophobic binding cavity of GOBP2 [32].

Next, we used homology modeling and molecular docking to further explore the molecular interaction between OachGOBPs and ligands. The constructed GOBP 3-D structures were verified by SAVES Server, which is a method with high reliability, stability, and consistency [33,34]. Ramachandran diagrams showed no amino acid residues in the disallowed regions, meeting high modeling standards (>90%). Validation 3-D was used to evaluate the compatibility of atomic models (3-D) with their own amino acids. The 3-D values of both OachGOBP1 and OachGOBP2 were 100%, indicating that the residual conformation of the constructed model is reasonable. The ERRAT scores were all greater than 85%, indicating that the non-bonding interactions between different atoms in the two models constructed were reasonable. The two OachGOBP proteins we constructed were reliable through the above three quality controls (Appendix A). Then, we calculated the binding energies of OachGOBPs and ligands (Table 1). The results showed that all the binding energies of OachGOBPs and ligands were negative, and the distances of all hydrogen bonds were less than 3 Å (Figure 6 and Figure 7), suggesting that there was a strong interaction between OachGOBPs and different ligands, similar to that of night moths [15] and *G. molesta* [27]. Previous studies have reported that insect OBPs usually bind different ligands in hydrophobic cavities [22,23,27,35,36,37]. Here, by analyzing binding models, we also found that some key hydrophobic residues were involved in the ligand binding of OachGOBP1 and OachGOBP2, suggesting that the binding mechanisms of OachGOBPs are similar to those identified in other insect GOBPs. For the number of amino acid residues within 4.0 Å surrounding the same ligand, GOBP2 is greater than GOBP1. OachGOBP2 shows a wider odorant binding spectrum and higher binding capacity than GOBP1, which is consistent with the results of fluorescence competitive binding experiments. In addition, we found some hydrophobic residues that bound to all ligands in both OachGOBP1 and OachGOBP2: 14 of OachGOBP1 (Val-31, Phe-35, Phe-56, Phe-59, Trp-60, Ile-75, Leu-84, Leu-85, Phe-99, Ile-117, Ile-134, Val-137, Ala-138 and Phe-141), 11 of OachGOBP2 (Phe-34, Phe-55, Phe-58, Trp-59, Ile-74, Leu-84, Ile-116, Val-133, Val-136, Ala-137, and Phe-140). In all, the result showed that some amino acid residues within the predicted binding pocket of selected OachGOBPs might be essential for the olfactory recognition of odorant cues from camphor plant volatiles and sex pheromone components. Therefore, these residues can be used as targets for further studies on the ligand binding mechanisms of OachGOBPs by integrating in vivo (CRISPR/Cas9 editing system) [31,38] and in vitro (site-directed mutation) [36,39].

## Figures and Tables

**Figure 1 insects-14-00216-f001:**
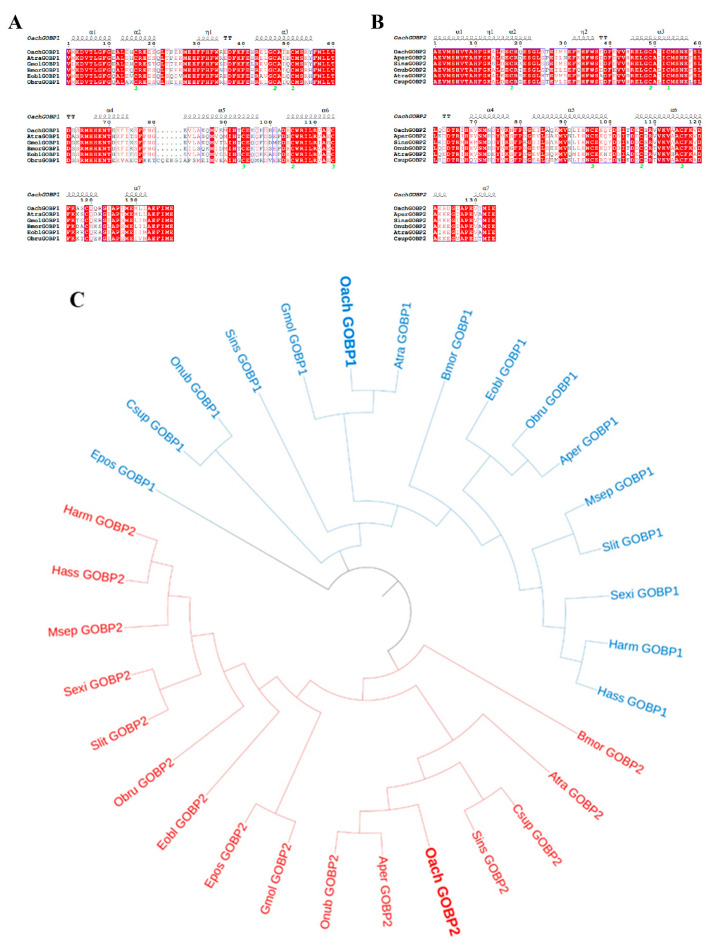
Sequence alignment and phylogenetic analysis of OachGOBPs. **A** and **B**: amino acid sequence alignment of OachGOBP1 and OachGOBP2 with their orthologs of lepidopteran insects. The disulfide bridges are numbered 1 to 3. The top line mark the position of alpha-helix (α1-7). Values at the nodes are the mean amino acid sequence length. The predicted conservative domain is represented in the blue box and 100% conserved residue is represented in the red shadow. **C**: phylogenetic analysis of OachGOBPs.

**Figure 2 insects-14-00216-f002:**
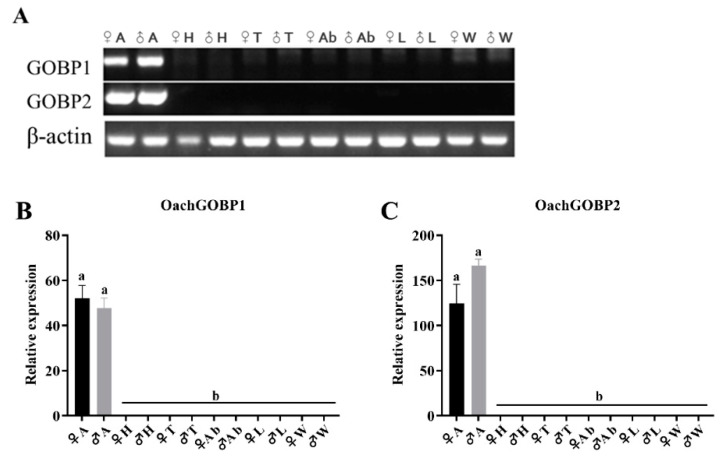
Expression profiles of OachGOBPs in *O. achatina*. (**A**): RT-PCR analysis of tissue expression of OachGOBPs with β-actin as reference gene. ♀, female; ♂, male. (**B**,**C**): qPCR analysis of relative expression (mean ± SE) in different tissues of OachGOBP1 and OachGOBP2, respectively. A: antennae; H: heads (with antennae and proboscises removed); T: thoraxes; Ab: abdomens; L: legs; W: wings.

**Figure 3 insects-14-00216-f003:**
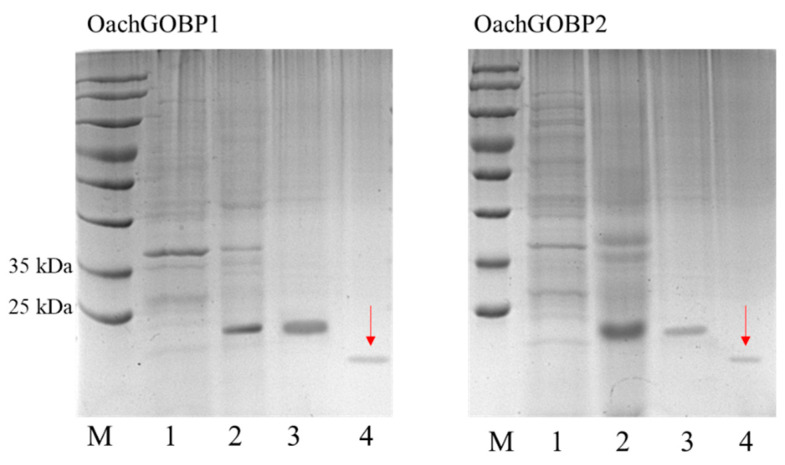
15% SDS-PAGE analysis of recombinant *pET*30a(+)/OachGOBPs protein. M: Protein molecular mass marker; 1: supernatant; 2: inclusion bodies; 3: recombinant *pET*30a(+)/OachGOBPs protein after purification with Ni-NTA; 4: target proteins without His-tag (indicated by red arrow).

**Figure 4 insects-14-00216-f004:**
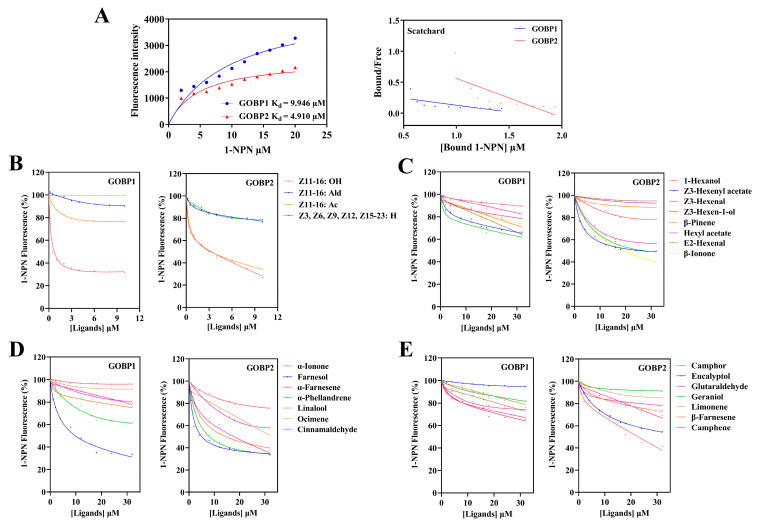
Binding of selected ligands to OachGOBPs. (**A**): Binding curves of 1-NPN to OachGOBP1 and OachGOBP2 with relative Scatchard plots. (**B**): Binding curves of OachGOBPs to 4 sex pheromone components. (**C**–**E**): Binding curves of OachGOBPs to 22 volatiles of camphor. The ligand names are shown on the right of the curves. The binding data of components were calculated and listed in Table 1.

**Figure 5 insects-14-00216-f005:**
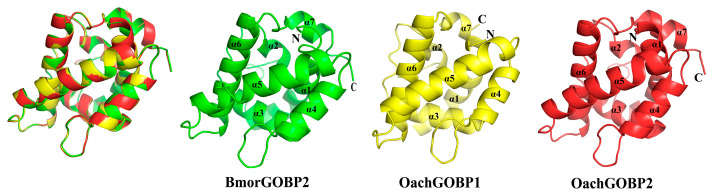
3-D Structural modeling of OachGOBPs. Superposition of the three GOBPs in the same orientation.

**Figure 6 insects-14-00216-f006:**
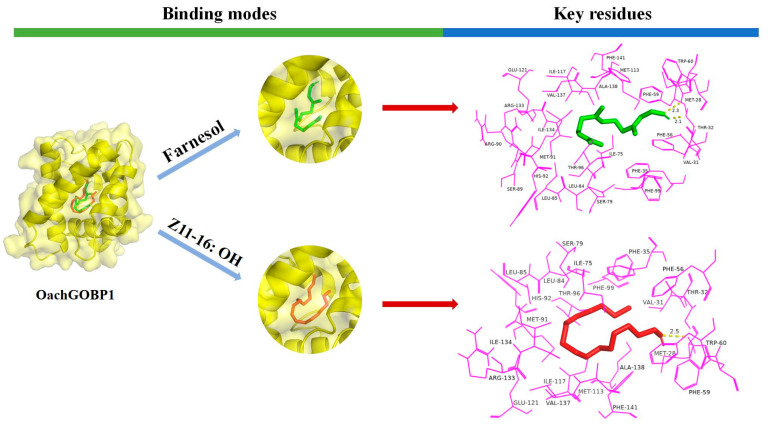
Binding modes and key residues of OachGOBP1 with different ligands: Farnesol (green) and *Z*11–16: OH (red) in the putative binding pocket of chain A of OachGOBP1. The key residues of the different ligands that interact with OachGOBP1. The residues within 4 Å of the ligands are highlighted in magenta. (For interpretation of the references to color in this figure legend, the reader is referred to the web version of this article.)

**Figure 7 insects-14-00216-f007:**
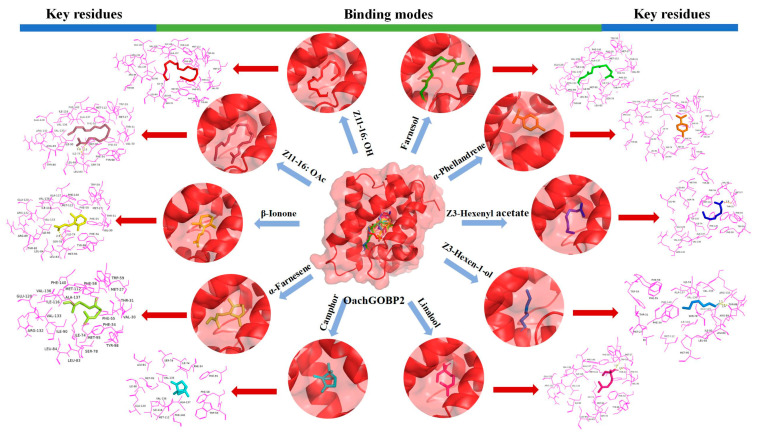
Binding modes and key residues of OachGOBP2 with different ligands: Farnesol (green), α-Phellandrene (orange), Z3-Hexenyl acetate (blue), Z3-Hexen-1-ol (marine), Linalool (hot), Camphor (cyan), α-Farnesene (limon), β-Ionone (yellow), *Z*11-16: OAc (raspberry) and *Z*11-16: OH (red) in the putative binding pocket of chain A of OachGOBP2. The key residues of the different ligands that interact with OachGOBP2. The residues within 4Å of the ligands are highlighted in magenta. (For interpretation of the references to color in this figure legend, the reader is referred to the web version of this article.)

**Table 1 insects-14-00216-t001:** Binding data of different ligands to OachGOBPs by binding assay and molecular docking.

Odor Compounds	GOBP1			GOBP2		
	IC_50_ (μM)	*K*_i_ (μM)	Free Binding Energy (kcal·mol^−1^)	IC_50_ (μM)	*K*_i_ (μM)	Free Binding Energy (kcal·mol^−1^)
Volatile compounds						
1-Hexanol	-	-	N	-	-	N
Z3-Hexenyl acetate	-	-	N	16.77	14.30	−5.4
Z3-Hexenal	-	-	N	-	-	N
Z3-Hexen-1-ol	-	-	N	21.36	18.18	−4.7
β-Pinene	-	-	N	-	-	N
Hexyl acetate	-	-	N	-	-	N
E2-Hexenal	-	-	N	-	-	N
β-Ionone	-	-	N	17.76	15.51	−8.0
α-Ionone	-	-	N	-	-	N
Farnesol	10.90	9.49	−7.5	8.68	7.33	−7.8
α-Farnesene	-	-	N	15.91	13.42	−7.7
α-Phellandrene	-	-	N	10.21	8.71	−7.2
Linalool	-	-	N	13.89	12.05	−6.1
Ocimene	-	-	N	-	-	N
Cinnamaldehyde	-	-	N	-	-	N
Camphor	-	-	N	20.72	17.49	−6.6
Eucalyptol	-	-	N	-	-	N
Glutaraldehyde	-	-	N	-	-	N
Geraniol	-	-	N	-	-	N
Limonene	-	-	N	-	-	N
β-Farnesene	-	-	N	-	-	N
Camphene	-	-	N	-	-	N
Pheromones compounds						
Z11-16: OH	1.79	1.57	−6.5	3.86	3.30	−6.7
Z11-16: OAc	-	-	N	3.30	2.84	−7.3
Z11-16: Ald	-	-	N	-	-	N
Z3, Z6, Z9,Z12,Z15–23: H	-	-	N	-	-	N

Notes: “-“ represent IC_50_ and *K*_i_ values of compounds without binding ability. *K*_i_ < 2 μM indicates high binding affinity, 2 μM < *K*_i_ < 5 μM indicates moderate binding affinity, 5 μM < *K*_i_ < 10 μM indicates low binding affinity for sex pheromones. *K*_i_ < 10 μM indicates high binding affinity, 10 μM < *K*_i_ < 20 μM indicates moderate binding affinity, 20 μM < *K*_i_ < 30 μM indicates low binding affinity, 30 μM < *K*_i_ indicates no binding capacity for plant volatiles. “N“ represents not tested.

## Data Availability

Data is contained within the article and Appendix A.

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
