# Peer review of "Identification and Functional Characterization of General Odorant Binding Proteins in *Orthaga achatina"

_insects, 2023, doi:10.3390/insects14030216_

Round 1
Reviewer 1 Report
Ma Y, Li Y et al. characterize the sequence and expression of 2 conserved GOBPs from a pest, O. achatina. Using in vitro and in silico approaches, they identified ligands for the 2 GOBP.
The study is well conducted; I only have minor comments:
1. introduction "In general [...] the expression of GOBP differs between female and male antennae. It is generally believed that the expression is higher in females than in males [7]" - The cited paper concluded that most (7 out of 9) antennal-enriched OBPs were expressed at similar levels between male and female.
2. Figure legends should include more information, for example:
Figure 1: description of the top line in panelA; indicate what the values at the nodes represent.
Figure 2: replicates for qPCR; statistic info missing
Table 1: add what "N" stands for
3. In the discussion, it is not clear why that author speculate that "OachGOBP2 has a similar function to AperGOBP2" and not similar function to AtraGOBP2 (most similar to OachOBP2 accroding to figure1).
4. It would be interesting to discuss the conservation of the "key hydro- 330 phobic residues" identified in this study.
Reviewer 2 Report
In this paper, authors have preposed General odorant binding protein should paly the crucial role in determing Odorant sensitivity and selectivity. You found that two full-length GOBP genes (OachGOBP1 and OachGOBP2) could be successfully cloned according to transcriptome sequencing results, and real time quantitative PCR measurements showed that both GOBP genes were specifically expressed in the antennae of both sexes, proposing their important roles in olfaction. I have checked out the series of reseraches carried out by authors. They are very interesting. However, to my knowledge, the underlying physical mechanism or the microscopic (quantum) process is still lacking until now. How is the relation of this GOBP gene with microscopic process or the enligitenment of this study for the mechanism of olfaction in microscopic aspect. I suggest these discussion should be added. overall, this reserach is very interesting and I recomand it to publish in this journal.
Author Response
Thank you for your valuable comments! Please check the attachment.

Reviewer 4 Report
The manuscript "Identification and Functional Characterization of General Odorant Binding Proteins in Orthaga achatina" by Yu Ma et al. presents the identification of two general odorant binding proteins (GOBPs) from the antennae of Orthaga achatina, their heterologous expression in E. coli and their binding affinity for some odorants and sex hormones. The work is complemented by phylogenetic analysis of OachGOBPs, homology modeling of the two GOBPs and molecular docking of odor and pheromone compounds.
Overall the work is scientifically sound and the methodology is presented clearly. The results are presented appropriately, however, the introduction can be significantly improved by adding more structural and biochemical information on GOBPs. Although the work is not innovative, the conclusions drawn are supported by the results obtained and thus could be of interest to the readers in the field.
I can suggest publication of the manuscript to Insects after the authors address the following minor issues.
Minor issues:
1. "KDa" in Fig. 3 should be "kDa" as in the text.
2. "Ki" in l. 211-217 and legend of Table 1 should be "Ki"
3. "(μM)" in Table 1 should be "(μΜ)"
4. "Binging modes" and "King residues" in Figs. 6 and 7 should probably read "Binding modes" and "Key residues".
5. The latter "A" in the parentheses, not brackets as indicated in the legend of Table S4 (Supplementary Material) should be replaced by "Å".
Author Response
Thank you for your valuable comments! We have revised the questions according to your suggestions and uploaded the manuscript.
Round 2
Reviewer 3 Report
The vast majority of my points have been addressed by the authors. I agree that the competition assay that they used is classic. Actually I also use this method in my laboratory. Nevertheless, my opinion is that this method has a lot of limitations. As an advice, I would be looking for another alternative or complementary “direct’ method e.g. ITC for evaluation of ligands’ binding.